# The Value of FET PET/CT in Recurrent Glioma with a Different IDH Mutation Status: The Relationship between Imaging and Molecular Biomarkers

**DOI:** 10.3390/ijms23126787

**Published:** 2022-06-17

**Authors:** Marija Skoblar Vidmar, Andrej Doma, Uroš Smrdel, Katarina Zevnik, Andrej Studen

**Affiliations:** 1Division of Radiotherapy, Institute of Oncology Ljubljana, 1000 Ljubljana, Slovenia; usmrdel@onko-i.si; 2Department of Nuclear Medicine, Institute of Oncology Ljubljana, 1000 Ljubljana, Slovenia; kzevnik@onko-i.si; 3Faculty of Medicine, University of Ljubljana, 1000 Ljubljana, Slovenia; 4Experimental Particle Physics Department, Jožef Stefan Institute, 1000 Ljubljana, Slovenia; andrej.studen@ijs.si; 5Faculty of Mathematics and Physics, University of Ljubljana, 1000 Ljubljana, Slovenia

**Keywords:** glioma, treatment-related changes, true progression, pseudoprogression, radiation necrosis, biomarkers, ^18^F-FET PET, IDH mutation

## Abstract

The evaluation of treatment response remains a challenge in glioma cases because the neuro oncological therapy can lead to the development of treatment-related changes (TRC) that mimic true progression (TP). Positron emission tomography (PET) using O-(2-[^18^F] fluoroethyl-)-L-tyrosine (^18^F-FET) has been shown to be a useful tool for detecting TRC and TP. We assessed the diagnostic performance of different ^18^F-FET PET segmentation approaches and different imaging biomarkers for differentiation between late TRC and TP in glioma patients. Isocitrate dehydrogenase (IDH) status was evaluated as a predictor of disease outcome. In our study, the proportion of TRC in IDH wild type (IDHwt) and IDH mutant (IDHm) subgroups was without significant difference. We found that the diagnostic value of static and dynamic biomarkers of ^18^F-FET PET for discrimination between TRC and TP depends on the IDH mutation status of the tumor. Dynamic ^18^F-FET PET acquisition proved helpful in the IDH wild type (IDHwt) subgroup, as opposed to the IDH mutant (IDHm) subgroup, providing an early indication to discontinue dynamic imaging in the IDHm subgroup.

## 1. Introduction

Molecular biomarkers have fundamentally changed the understanding of glioma over the last decade. Accordingly, the fifth edition of the World Health Organization Classification of Tumors of the Central Nervous System (WHO CNS5) incorporates numerous molecular biomarkers with clinicopathologic utility that are important for more accurate classification of CNS neoplasms [1]. Molecular biomarkers also improve diagnostic accuracy and influence the course of treatment by changing treatment recommendations [2]. A marker of particular importance is isocitrate dehidrogenase (IDH). Mutations in genes encoding IDH are known to play a crucial role in the classification of gliomas. IDHm glioma generally exhibits a better disease outcome than IDHwt. The IDHm is an independent predictor of prolonged survival and its prevalence is inversely correlated with tumor grades [3,4]. In adults, diffuse gliomas have been divided into three types according to the new classification: (1) astrocytoma, IDHm; (2) oligodendroglioma, IDHm and 1p/19q-codeleted; and (3) glioblastoma, IDHwt. Prior to that, glioblastomas were diagnosed based on the histologic findings including both IDHm and IDHwt tumors with very different biological features and prognoses. In WHO CNS5, glioblastomas comprise only IDHwt tumors. In addition, IDHwt diffuse astrocytic tumors in adults without the histologic features of glioblastoma but having one or more of three genetic parameters (TERT promoter mutation, EGFR gene amplification, combined gain of entire chromosome 7 and loss of entire chromosome 10) are classified as glioblastomas. In the new classification, all IDHm diffuse astrocytic tumors are considered a single type astrocytoma, IDHm and are graded as gradus 2, 3, or 4. Grading of gliomas also takes into account some other molecular findings such as the presence of CDKN2A/B homozygous deletion, which results in a worse prognosis and is subsequently graded as a WHO grade 4. The separation into IDH wild type and mutant tumors is an important advancement and a key factor in the treatment, follow-up, and understanding of glial tumors [5].

The treatment of gliomas includes maximal surgical resection, possibly followed by radiotherapy (RT) and chemotherapy with either procarbazine/lomustine/vincristine (PCV) or temozolomide (TMZ). Due to the proliferative, radioresistant, and chemoresistant nature of the gliomas and high levels of intratumoral heterogeneity, the disease often recurs, and the possibilities of additional treatment are very limited [6].

In the regular clinical work, the glioma treatment response assessment is based on imaging diagnostics, primarily MRI. MRI is the mainstay of imaging gliomas to monitor both treatment and response. T1-weighted MRI without and with contrast medium, T2-weighted as well as fluid-attenuated inversion recovery (FLAIR) MRI sequences are used for anatomic imaging [7,8,9].

Irradiation of brain tumors causes damage to the blood-brain barrier, which can lead to extravascular leakage of the contrast medium, which may have the same appearance on magnetic resonance imaging (MRI) images as a vital residual tumor or TP. This side effect of oncology treatment is defined as pseudoprogression and begins to occur approximately three months after irradiation, with the incidence of up to 50% in combined oncology treatment. Another radiation-induced side effect is radionecrosis (RN), possibly due to vascular injury and glial cell damage, usually occurring six months after the irradiation. RN can have the appearance of tumour residue or tumor recurrence on postcontrast MRI. The incidence of RN is estimated at up to 30% and increases with the length of the time from irradiation [10,11].

TRC, such as pseudoprogression and RN, overlaps with TP. This makes the differentiation challenging, and can consequently complicate the treatment course and compromise care. Therefore, the correct differentiation between TRC and actual TP continues to be a crucial issue [12,13]. For these reasons, additional imaging methods such as perfusion MRI or MR spectroscopy and functional methods such as ^18^F-FET PET are used. ^18^F-FET PET CT is based on the evaluation of transport of ^18^F labeled tyrosine in tissues. In gliomas, ^18^F-FET uptake significantly correlates with tumor cell density and neoangiogenesis, all biological hallmarks of highly malignant glial tumors. [14,15,16]. The aim of this study was to assess the diagnostic performance of different ^18^F-FET PET segmentation approaches for differentiation between late TRC and TP in glioma patients with different IDH mutation statuses.

Since we investigated the late effects of radiochemotherapy, TRC was associated with radiation necrosis in our study.

## 2. Materials and Methods

### 2.1. Subjects

This retrospective study included 47 patients who were treated at the Division of Radiotherapy, Institute of Oncology in Ljubljana and, on the recommendation of the multidisciplinary tumor board, were referred to our Nuclear Medicine Department for ^18^F -FET PET imaging between April 2019 and October 2021 in order to distinguish between TP and TRC. All patients who had undergone a standard MRI were able to understand the reason for additional ^18^F-FET PET imaging. All patients had previously been diagnosed with adult diffuse gliomas and had a prior biopsy and radiochemotherapy according to EANO guidelines. All patients had a prior MRI suspicious of TP, as determined by the Response Assessment in Neuro-Oncology (RANO) working group criteria.

The study was approved by the institutional review board committee (approval number ERIDNPVO-0073/2021). All involved persons gave their written informed consent prior to study inclusion. The study conformed to the ethical norms and standards in the Declaration of Helsinki. All biological material was administered according to international guidelines.

### 2.2. Determination of IDH Genotype

The IDH mutation status was assessed by the IDH1R132H protein expression level evaluated by immunohistochemistry until early 2017 (15 pts), and after that using Next Generation Sequencing of a Glioma-Tailored Gene Panel (29 pts). For ^18^F-FET PET analysis, patients were split into IDHm and IDHwt groups. Next-generation sequencing (NGS) is being increasingly used in routine clinical practice, including for the diagnostics of rare entities like gliomas because it can replace multiple single-gene genomic testing technologies while requiring only one test. Gene-targeted NGS offers a cost-effective approach to simultaneous detection of multiple genetic alterations with a minimal amount of sampled DNA while achieving high sensitivity. This makes this method highly attractive for use in gliomas. Specifically designed panels for gliomas are needed for the routine diagnosis of these tumors. We use the isolation of DNA from FFPE tissue using the Maxwell RSC FFPE Plus DNA Purification kit (Promega). The NGS panel assesses mutations in specific target regions (“Hotspots”) in 9 genes: BRAF, H3F3A, HIST1H3B, HIST1H3C, IDH1, IDH2, KRAS, NRAS, pTERT, analysis of the entire coding region of 11 genes: ACVR1, ATRX, CIC, FUBP1, EGFR, FGFR1, PIK3CA, PIK3R1, PTEN, SETD2, TP53,12 gene copy number (CNV) analysis: CDKN2A, CDKN2B, EGFR, FGFR1, MDM2, MDM4, MET, MYCN, PDGFRA, PIK3CA, PIK3R1, PTEN and chromosome level analysis. Analysis of results is performed with the IonReporter software package (Thermo Fisher Scientific) (reference genome hg19). Border detection (default filter) is set to 5.0% (mutation rate relative to unmutated DNA). The sensitivity of the method is 99.21% (hotspot), 96.88% (indel), 97.10% (de novoSNV), 85.71% (de novo indel), and 95.35% (fusion). Negative results (no mutations) do not exclude the presence of mutations, amplifications, or deletions below the limit of detection [17].

### 2.3. ^18^F-FET PET Imaging

The synthesis of ^18^F-FET was performed by IASON GmbH (Graz, Austria). We used an integrated PET/CT system (Biograph mCT 64; Siemens, Erlangen, Germany) for acquisition of dynamic PET images over 40 min, starting immediately after injection of 3MBq of ^18^F-FET per kg of body weight. All patients fasted for at least 6 h prior to PET acquisition. Dynamic 40-min scans were acquired using 35 sequences (200 × 200 matrix; 12 × 5 sec; 6 × 10 s; 6 × 30 s; 5 × 60 s; 6 × 5 min). PET images were reconstructed with ordered- subset expectation maximization (OSEM) algorithm using 2 iterations with 21 subsets and gauss filtering to a full width at a half maximum (FWHM) of 5mm. High-resolution static images (400 × 400 matrix) were reconstructed from 20–40 min post-injection scans with OSEM algorithm consisting of two iterations with 21 subsets and Gaussian filtering to a FWHM of 3 mm.

### 2.4. ^18^F-FET PET Image Analysis

PET scans were interpreted by two experienced nuclear medicine physicians, who were blinded to the histological and clinical data. The assessment of the tumor maximal standardized uptake value (SUV_max_) was performed for each lesion by placing a spherical Volume-of-Interest (VOI) over the area of maximal amino acid uptake in the tumor on summed 20–40 min post-injection PET images. The mean standardized uptake value of the normal background brain tissue (SUV_mean_bg_) was determined by placing a crescent-shaped VOI over the contralateral, unaffected hemisphere including white and grey matter. A tumor volume segmentation using a 3-dimensional auto contouring process with a tumor-to-background ratio (TBR) cutoff of at least 1.6 g/mL was used to determine the mean standardized uptake value of the tumor (SUV_mean_tumor_). This cutoff was based on the results of a biopsy-controlled study in which a lesion-to-brain ratio of 1.6 g/mL resulted in the best separation between tumor and non-tumor tissue [16]. Respective ratios of the TBR_mean_ and TBR_max_ were calculated by dividing SUV_mean_tumor_ and SUV_max_, respectively, by the SUV_mean_bg_. For Time Activity Curve (TAC) evaluation, a spherical VOI was centered over the area of the highest tumor uptake to the entire dynamic datasets. Time-to-Peak (TTP) was determined as a time (in minutes) from the beginning of the dynamic PET acquisition up to the peak activity in the lesion. In lesions with constantly increasing TAC without an identifiable peak, TTP was defined as the end of the dynamic PET acquisition. We identified a cut-off point at 22.5 min post-injection as a dip in a distinct two-peak distribution of patients with respect to TTP with an early group reaching a distinct maximum before the prescribed threshold and a late group reaching it later or not exhibiting a peak at all. Using the cut-off, we identified the following curve categories, in order of increasing shape score:TAC score of −1: lesions with an early peak in SUV, followed by a constant descent of activity;TAC score of 0: lesions with ascending SUV reaching an early peak before 22.5 min, followed by a plateau or small descent of less than 5%;TAC score of 1: lesions with constantly increasing SUV without an identifiable peak.

### 2.5. Diagnosis of TP

Diagnosis of TP was based on histopathologic analysis following surgery, by clinical deterioration, and/or further radiological progression in a follow-up MRI at least four weeks after the initial assessment. In contrast, the diagnosis of TRC was applied in cases of negative histopathology, stable clinical conditions (with no treatment changes within the follow-up time), or stabilization/regression of the contrast-enhancing lesions at follow-up MRI (at least four weeks following initial assessment), respectively. The diagnosis for some patients was confirmed by more than one modality. Thus, the classification criteria in our study were similar to those of previous investigations [18,19].

### 2.6. Statistical Analysis

Descriptive statistics were used to calculate typical measures in patients’ demographic and clinical characteristics. Data were expressed as median with a range, and categorical data were expressed as counts and frequencies. Statistical analyses were carried out using IBM SPSS Statistics software version 26 (Statistical package for the Social Sciences Statistical Software; SPSS Inc, IBM Corporation, Armonk, NY, USA). 

We analyzed the FET outcome data using R statistical software (R version 3.1.1 (2014-07-10), R Foundation for Statistical Computing, Vienna, Austria, https://www.R-project.org/ (accessed on 16 June 2022). To statistically assess significant differences in ROC curves we performed a non-parametric ROC analysis [20] We used the optimal operating point on the ROC curve using the Youden index, and uncertainty in cut-off values was modelled using a large number approach [21]. We compared patient groups using the Mann-Whitney-Wilcoxon rank-sum U test [22]. We combined TAC score, IDH mutation status, and TBR_mean_ predictions using the logistic regression (LR) model. When used on patients grouped by IDH mutation status, we only used TAC score and TBR_mean_ variables in the LR model. Results with a *p*-value below 0.05 were deemed statistically significant, and 95% confidence intervals (CI_95%_) were used to quantify uncertainty in statistically derived values.

## 3. Results

We analyzed the data of 47 patients with glial tumors who underwent ^18^F-FET PET for differentiation between TP and TRC. The interval between the end of radiation therapy and subsequent PET imaging was no less than 12 weeks in all cases.

Forty-four patients were eligible for analysis, and their median age was 44 (17 to 72, SD 14 years). Twenty-seven (61.4%) were male and 17 (38.6%) were female.

IDHm and IDHwt were present in 26 (59.1%) and 18 (40.9%) patients, respectively.

TP and TRC were confirmed in 32 (72.7%) and 12 (17.7%), respectively. The proportion of TRC in IDHwt 5/18 (27.8%) and IDHm 7/26 (26.9%) groups, was without significant difference (*p* = 0.61).

Diagnosis of TP/TRC was confirmed through histopathological analysis following surgery, through MRI, or based on clinical deterioration. 

Regarding the verification of the diagnosis, 11 patients had surgery, 38 patients got a confirmed diagnosis after repeated MRI and 34 patients experienced clinical deterioration. Two out of 12 patients with TRC (2/12) had surgery, as opposed to nine patients with true progression (9/32). The proportion of those operated on is not significantly different between these groups.

The time from diagnosis to ^18^F-FET PET was 104 weeks (84 weeks in IDHwt and 130 weeks in IDHm; the difference in median time to evaluation was without significance regarding the TRC or TP (*p* = 0.5), as well as IDH status (*p* = 0.9).

The proportion of patients with TRC according to IDH status was not significantly different (χ^2^
*p* = 0.9).

In this group of patients, the overall survival was excellent, with median survival exceeding 500 weeks. The median survival according to IDH status was not different, though the median survival was not reached in the TRC group, the analysis is underpowered to detect significance.

^18^F-FET-PET parameters were then analyzed for the whole group and selectively according to different IDH mutation statuses. For the whole group, the SUV_max_ value had a mean of 4.04 and a median of 3.78, with the SD of 1.83 (IDHm: 4.10, 3.82, and 1.73, IDHwt: 4.02, 3.76, and 2.05, respectively). The difference between groups was not statistically significant. 

When comparing the mean values for SUV parameters according to the IDH mutation status, we found that while in IDHwt patients there are no significant differences in SUV values according to radio necrosis and progression, in the IDHm, the mean values of SUV_max_ (TRC/TP (p): 4.7/2.49 (0.001)), TBR_max_ and TBR_mean_ differed significantly (Table 1).

Figure 1 shows a comparison of ROC curves for different FET- derived variables split by IDH mutation status. For all patients, TBR_max_ and TBR_mean_ together with logReg show a statistically significantly better performance compared to TTP. In groups split by IDH status, no statistically significant differences in ROC curves could be determined. Relapse was associated with larger values of TBR_max_ and TBR_mean_, while the opposite association to lower values was identified for TTP. In IDHm, the TTP ROC is below the diagonal, indicating an inverse relationship between higher TTP and relapse, which is not statistically confirmed. In IDHwt cases, TTP is the best predictor, its difference to either TBR_max_ or TBR_mean_ is, however, not significant. We found a single significant variable, TBR_mean_, with a coefficient of 1.6 (*p* = 0.03) and equivalent odds ratio of 4.9 (CI_95%_: 2.1–8.5) per unit change in TBR_mean_ in the LR model for the full patient group. We identified no significant variables in IDH-specific LR models.

Table 1 shows the ability of FET in predicting the tumor status. For each IDH group we show the predictive quality of FET variables -TBR_max_, TBR_mean_, TTP and LR. For each variable we show its mean and range in the TRC group, mean and range in TP group, optimum cut-off based on the Youden index, sensitivity and specificity with associated 95% confidence intervals at cut-off point and the *p*-value associated with the MWU test for patients grouped by tumor outcome. The variables that can identify tumor outcome with statistical significance are shown in bold. Two regimes can be identified: in the full and IDHm group, TBR_max_ and TBR_mean_ are significant and TTP is irrelevant. Conversely, TTP becomes significant and TBR_mean_ and TBR_max_ irrelevant in the IDHwt group. In all cases the LR model which combines TBRmax and TAC score is a significant predictor of the tumor outcome.

## 4. Discussion

This is the first reported study of performance of ^18^F -FET PET for the differentiation between TP and TRC in glioma patients based on IDH mutation status. We evaluated the diagnostic potential of static and dynamic ^18^F -FET PET parameters for differentiation between late TRC and TP based on different IDH mutation subgroups in mixed diffuse glioma patients. The differentiation between TP and TRC represents one of the most frequent indications for the use of amino acid PET in clinical practice. PET using radiolabeled amino acids is gaining increasing interest for the diagnostics of brain tumors because a conventional MRI is limited in differentiating tumor tissue from nonspecific tissue changes following neuro-oncological treatment. Recently, the RANO working group has recommended the additional use of amino acid PET imaging for brain tumor management [23].

There is a growing body of evidence that IDH mutations play a role in the formation of brain tumors and influence the response to neuro oncological treatment and overall survival [24,25]. *IDH* mutations in glioma are associated with significantly prolonged progression-free and overall survival compared with IDHwt tumors. There is now a greater appreciation that the biology of IDHm glioma is quite different from that of IDHwt tumors and that tumorigenic processes most likely are different as well. While the specific mechanism of IDH mutation that results in the oncogenic switch in gliomas remains unknown, potential mechanisms have been identified, including the inhibition of hypoxia-related proline hydroxylases, inhibition of DNA demethylases, inhibition of histone demethylases, and alterations in glutamate metabolism. Further work is needed to elucidate the specific role of IDH mutation and the pathological consequences that clearly affect tumor evolution and prognosis. Given the complex role of the IDH mutation in the progression, aggressive biological behavior, and response to the treatment of diffuse gliomas, it seems reasonable to observe and analyze these two molecular groups separately [26,27,28,29,30]. 

In our study, the static ^18^F-FET PET measures TBRmax and TBRmean outperformed the dynamic parameters in the IDH (all) group (accuracy 79 and 76% respectively, *p* = 0.001 both) and the IDHm group (accuracy 91 and 87%, p of 0.004 and 0.01, respectively); dynamic measure TTP only achieved an accuracy of 60 and 61% in IDH (all) and the IDHm group, respectively, and the p value of the MWU test was not significant. The situation was reversed in the IDHwt group, where TTP was found to be a significant predictor (accuracy of 83%, *p* = 0.05) while static measures showed a mediocre performance (accuracy of 67 and 72% respectively, non-significant p values of MWU test).

The results of our study contradict several previous reports, where authors described improved diagnostic sensitivity when dynamic metrics were added to static ^18^F-FET metrics alone. 

The difference in the results reported could be due to distinctly different patient populations: while previous studies [31,32,33,34,35] did not take into account the IDH mutation status of the participants, IDH mutation was present in 59.1% of our patients, this being a high proportion compared to usual rates in glioma patients. IDH mutation rates vary substantially between different types of glioma, nonetheless they are not common [28,29].

Therefore, the significant diagnostic performance of dynamic PET acquisition in a non-divided group of glioma patients, the majority of which would typically be IDHwt, was impaired by a large IDHm subgroup included in our study.

Due to the limited availability of the radiotracer at our institution we consider performing ^18^F-FET PET/CT investigation particularly when MRI yields inconclusive results between TRC and TP. Important for our study is that ^18^F-FET PET/CT imaging was considered appropriate only if it resulted in therapeutic consequences. The patients with poor performance status and without further treatment options, typically being IDHwt, were therefore following the standard of care not assigned to receive ^18^F-FET PET imaging, thus further reducing the number of IDHwt subjects in our study. Therefore, a higher proportion of IDH mutations in our study is a consequence of more frequent equivocal decisions at a multidisciplinary tumor board in IDHm cases, and represents a selection of notably difficult cases [31]. A high rate of IDHm in our patients also enabled this IDH mutation status-based study.

In comparison to the static acquisition, the use of dynamic acquisition of ^18^F-FET PET is very time-consuming and can therefore be challenging in an otherwise busy imaging department. Dynamic imaging incurs additional costs as a consequence of the prolonged decay of the tracer intended for consecutive patients, and fewer scans are performed within normal working hours, hence shorter imaging times are preferred. In our study, the shape of the dynamic curve, in general, has not been statistically significantly associated with the TRC or TP, although there was a trend that associated the plateau-shaped curve with the progressive disease. Our results suggest that the exclusion of dynamic acquisition and the performance of only static acquisition in IDHm patients could be a cost-effective strategy without the diagnostic potential of the investigation being hampered. Nonetheless, since dynamic imaging metrics proved useful in the IDHwt subgroup, these patients should be provided with the dynamic acquisition. [32,33].

While the cutoff values of TBR_mean_ in the all, IDHm and IDHwt subgroups (2.04, CI_95%_:1.8–2.3; 1.96, CI_95%_:1.7–2.3; and 2.09, CI_95%_:1.6–2.6, respectively) as determined in our study, within CL agree with previously reported cutoff values of 1.9–2.0 for the differentiation of both early and late TRC from true progression, the cutoff values of TBR_max_ for both all and IDHm (3.03, CI_95%_:2.6–3.4), and IDHwt (2.9, CI_95%_: 2.2–3.6) subgroups, as determined in our study, were above the values, reported by the majority of authors [31,32,33,34]. An optimal TBR_max_ cutoff value of about 1.9 with an accuracy of 85% was determined in late TRC glioblastoma patients, while a TBR_max_ cutoff of 2.3 (accuracy 96%) was determined in the early glioblastoma pseudoprogression by Galldiks et al. [18,34,35]. However, Kartels et al. estimated the optimal TBR_max_ cutoff value to be 3.52 in a late glioblastoma multiforme group of patients, indicating non-uniform outcomes in different patient group settings [36]. The reason for only moderate accuracy in differentiating TP from TRC might be due to a non-homogeneous group of patients because we did not limit our study to high-grade gliomas or specific treatment regimens. As reported in a meta-analysis by Cui et al., the accuracy of FET is known to be higher in high-grade glioma than in the mixed glioma patients group [37].

## 5. Conclusions

Differentiating TRC from TP is of critical importance for patient management and prognosis and it can often be challenging. In our study, the proportion of TRC in IDHwt and IDHm subgroups was without significant difference. We found that the diagnostic value of static and dynamic biomarkers of ^18^F-FET PET for discrimination between TRC and TP depends on the IDH mutation status of the tumor. Dynamic biomarkers play an important role in the IDHwt subgroup, and as opposed to the case of IDHm, the dynamic acquisition of ^18^F-FET PET might eventually be discontinued. Further prospective research in large sample sizes is needed to determine the value of ^18^F-FET PET in different molecular biomarker settings and to confirm our findings.

## Figures and Tables

**Figure 1 ijms-23-06787-f001:**
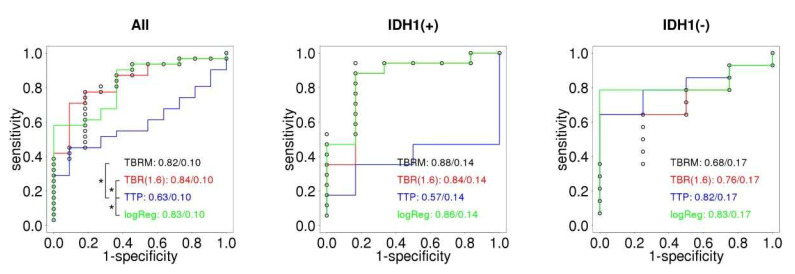
Receiver–operator characteristic (ROC) curves in classifying tumor outcome based on FET-derived parameters for different IDH mutation status groups. All: all patients in the study. IDH1(m): patients with mutated IDH gene. IDH1(wt): patients with IDH wild type. TBRM is TBR_max_, TBR(1.6) is TBR_mean_ at SUV 1.6 g/mL cutoff, TTP is time to peak and log Reg is the logistic regression model. The image shows AUC values with associated standard deviation. The asterisk associated brackets identify variables with statistically significant ROC curves as evaluated by a non-parametric significance test.

**Table 1 ijms-23-06787-t001:** Analysis of sensitivities, specificities, thresholds and diagnostic accuracy of ^18^F-FET PET biomarkers in differentiation between late TP and TRC in glioma patients.

**All** (*N* = 42)				**IDHm** (*N* = 23)				**IDHwt** (*N* = 18)			
**TBR**		**TTP**		**TBR**		**TTP**		**TBR**		**TTP**	
Max	Mean	(min)	LR	Max	Mean	(min)	LR	Max	Mean	(min)	LR
TP (count, mean, median, range)											
*N* = 31				*N* = 17				*N* = 14			
4.1	2.2	26	0.8	4.2	2.2	30	0.8	4.0	2.1	22	0.8
4.0	2.1	32		4.1	2.1	40		3.8	2.2	14.5	
1.1–8.0	0–3.2	5–40	0.1–1	2.1–6.4	1.7–3.2	7–40	0.3–1	1.1–8.0	0–3.1	7–40	0.4–1
**TRC** (mean, median, range)											
*N* = 11				*N* = 6				*N* = 4			
2.6	1.5	35	0.5	2.6	1.6	30	0.5	2.7	1.4	40	0.6
2.3	1.9	40		2.2	1.8	32		2.6	1.9	40	
1.6–4.2	0–2.2	12–40	0.1–0.8	1.9–4.1	0–2.2	12–40	0–0.9	1.6–4.2	0–2.0	40–40	0.4–0.6
**Threshold** (optimum, CI_95%_)											
3.03	2.04	32	0.79	3.03	1.96	32	0.66	2.9	2.09	40	0.65
2.6–3.4	1.8–2.3	28–36	0.7–0.9	2.6–3.4	1.7–2.3	27–37	0.6–0.8	2.2–3.6	1.6–2.6	36–40	0.6–0.8
**Sensitivity** (%, value at optimum, CI_95%_)											
77	71	48	58	94	88	83	88	64	64	79	79
60–89	53–84	32–65	41–74	73–99	66–97	54–97	66–97	39–84	39–84	52–92	52–92
**Specificity** (%, value at optimum, CI_95%_)											
82	91	91	100	83	83	53	83	75	100	100	100
52–92	62–98	62–98	74–100	44–97	44–97	31–74	44–97	30–95	51–100	51–100	51–100
**Accuracy** (%, value at optimum, CI_95%_)											
79	76	60	69	91	87	61	87	67	72	83	83
64–88	61–87	44–73	54–81	73–98	68–95	41–78	68–95	44–84	49–88	61–94	61–94
***p*-value**											
0.001	0.001	0.18	0.002	0.004	0.01	0.61	0.01	0.33	0.14	0.05	0.05

## Data Availability

The data presented in this study are available on request from the corresponding author. The patient data are not publicly available due to ethical restrictions.

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
