# Peer review of "The Value of FET PET/CT in Recurrent Glioma with a Different IDH Mutation Status: The Relationship between Imaging and Molecular Biomarkers"

_ijms, 2022, doi:10.3390/ijms23126787_

Round 1

Reviewer 1 Report

The manuscript "The value of FET PET/CT in recurrent glioma with a different IDH mutation status: the relationship between imaging and molecular biomarkers” is interesting and relevant in the complicated follow-up of Glial tumors; it shows results using high technology analysis (PET, genomic studies and sophisticated mathematical methods) in a well-designed study; some minor points could be addressed:

1.       Characteristics and relevance of IDH in Glial tumors should be briefly described in Introduction (L32-34 do not mention its genetic importance, nor the pathological consequences of mutations in brain neoplasms). Since IDH mutations are at the core of this research this point must be clarified.

2.       The use of pleonasms in cancer therapy is not justified (e.g. "was excellent” L170).

3.       At the clinical follow-up differences between TP and TRC are difficult in some cases, this circumstance could be briefly addressed in “Discussion”.

Author Response

Dear reviewer,

First of all, I would like to thank you for your helpful comments and suggestions.

Point 1:   Characteristics and relevance of IDH in Glial tumors should be briefly described in Introduction (L32-34 do not mention its genetic importance, nor the pathological consequences of mutations in brain neoplasms). Since IDH mutations are at the core of this research this point must be clarified.

Response 1:

In adults, diffuse gliomas according to new classification have been divided into three types:1) astrocytoma, IDHm; 2) oligodendroglioma, IDHm and 1p/19q-codeleted; and 3) glioblastoma, IDHwt. Until then glioblastomas were diagnosed based on the histologic findings including both IDHm and IDHwt tumors with very different biologies and prognoses. In WHO CNS5, glioblastomas will comprise only IDHwt tumors. In addition, IDHwt diffuse astrocytic tumors in adults without the histologic features of glioblastoma but having one or more of three genetic parameters (TERT promoter mutation,EGFR gene amplification, combined gain of entire chromosome 7 and loss of entire chromosome 10) will also be classified as glioblastomas. In the new classification, all IDHm diffuse astrocytic tumors are considered a single type astrocytoma, IDHm  and are graded as 2, 3, or 4. Grading of these tumors will also take into account other molecular findings such as the presence of CDKN2A/B homozygous deletion, which results in a worse prognosis and which will now have a WHO grade of 4. This separation into IDH wild type and mutant tumors is an important advancement and a key factor in the treatment, follow-up, and understanding of glial tumors.

Point2: The use of pleonasms in cancer therapy is not justified (e.g. "was excellent” L170).

Response 2:

We totally agree with you, so we rephrased the sentence: In this group of patients, the median survival exceeded 500 weeks.

Point3: At the clinical follow-up differences between TP and TRC are difficult in some cases, this circumstance could be briefly addressed in “Discussion”.

Response 3:

In our study, TRC was related to late post-radiation changes (more than 6 months after radiotherapy) or radiation necrosis.

Reviewer 2 Report

The manuscript entitled:" The value of FET PET/CT in recurrent glioma with a different  IDH mutation status: the relationship between imaging and  molecular biomarkers" focused on the evaluation of mutational status of most clinically relevant biomarkers and their relationship with imaging data to predict clinical outcome in glioma patients requires several moderate integrations to be accepted for the publication on this journal:

- In the introduction section, please, could the authors better explain the achronymous for TP and TRC?

- In the material and methods section (lines 68 -75) please, could the authors also report the incluision criteria followed to enroll patients in this study? In addition, could the authors report if biological material was administrated according to international guideliens?

- In the material and methods section (lines 76-80), the authors describe the conventional technology adopted to evaluate ID1 and IDH2 mutational status. Accordingly, NGS approach is not adequately discussed. Please, could the authors report technical details (reference range, sensitivity, specificity, limit of detection) usefull to accept positive status in IDH1/2 molecular analysis?

- In the results section, the authros elucidate positive status for IDH1/2 in enrolled patients. As regards, please, could the authors evaluate if sttistically relevant differences may be observed according to the technology? In addition, it is anachronystic only consider IDH1/2 positive status. In the recent era, the qualification of detected mutation is essential for the clinical purposes. Please, could the authors highlight how this data impact on the clinical results?

- In the results section, please, could the authors analyze clinical data at the sight of pathologic classiffication of enrolled patients( grading, histological diagnosis)?

Author Response

Dear reviewer,

First of all, I would like to thank you for your helpful comments and suggestions.

Point1: In the introduction section, please, could the authors better explain the achronymous for TP and TRC?

Response1:

TRC encompasses two situations, and these are pseudoprogression and radiation necrosis. Pseudoprogression describes the phenomenon that, in the absence of actual tumor growth, the diameter of contrast-enhancing areas enlarges more than 25% or new lesions occur during or after therapy, mimicking tumor progression within the first three months after chemoradiation completion with subsequent improvement of MRI findings. Radiation necrosis denotes an injury of brain tissue that is related to irradiation and may occur several months or even years after radiotherapy completion. Since we investigated the late effects of radiochemotherapy, TRC was associated with radiation necrosis in our study.

Point2: In the material and methods section (lines 68 -75) please, could the authors also report the incluision criteria followed to enroll patients in this study? In addition, could the authors report if biological material was administrated according to international guideliens?

Response2:

This retrospective study included 47 patients who were treated at the Division of Radiotherapy, Institute of Oncology in Ljubljana and, on the recommendation of the multidisciplinary tumor board and in order to distinguish between TP and TRC, were referred to our Nuclear Medicine Department for 18F-FET PET imaging between April 2019 and October 2021. All patients who had undergone standard MRI were able to understand the reason for additional 18F-FET PET imaging. All patients had previously been diagnosed with adult diffuse gliomas and had a prior biopsy and radiochemotherapy according to EANO guidelines.

The study was approved by the institutional review board committee (approval number ERIDNPVO-0073/2021) and biological material was administered according to international guidelines.

Point3: In the material and methods section (lines 76-80), the authors describe the conventional technology adopted to evaluate ID1 and IDH2 mutational status. Accordingly, NGS approach is not adequately discussed. Please, could the authors report technical details (reference range, sensitivity, specificity, limit of detection) usefull to accept positive status in IDH1/2 molecular analysis?

Response3:

Next-generation sequencing (NGS) is increasingly being used in routine clinical practice, including for the diagnosis of rare entities like gliomas because it can replace multiple single-gene genomic testing technologies while requiring only one test. Gene-targeted NGS offers a cost-effective approach to simultaneously detect multiple genetic alterations with a minimal amount of DNA and with high sensitivity, which makes this method highly attractive for use in gliomas. Panels that are specifically designed for gliomas are needed for the routine diagnosis of these tumors. We use the isolation of DNA from FFPE tissue using the Maxwell RSC FFPE Plus DNA Purification kit (Promega). New generation target sequencing (NGS, Ion S5, AmpliSeq Glioma Custom Design, Thermo Fisher Scientific) at the level DNA: analysis of somatic mutations in specific target regions ("Hotspots") in 9 genes:BRAF, H3F3A, HIST1H3B, HIST1H3C, IDH1, IDH2, KRAS, NRAS, pTERT, analysis of the entire coding region of 11 genes:ACVR1, ATRX, CIC, FUBP1, EGFR, FGFR1, PIK3CA, PIK3R1, PTEN, SETD2, TP53,12 gene copy number (CNV) analysis:CDKN2A, CDKN2B, EGFR, FGFR1, MDM2, MDM4, MET, MYCN, PDGFRA, PIK3CA, PIK3R1, PTEN and chromosome level analysis: chromosome lever 1p, 10q and 19q and chromosome 7. Analysis of results using the IonReporter software package (Thermo Fisher Scientific) (reference genome hg19). Border detection (default filter) is set to 5.0% (mutation rate relative to unmutated DNA). Negative result (no mutations) does not exclude the presence of mutations, amplifications or deletions below the limit of detection.

Point 4: In the results section, the authros elucidate positive status for IDH1/2 in enrolled patients. As regards, please, could the authors evaluate if sttistically relevant differences may be observed according to the technology? In addition, it is anachronystic only consider IDH1/2 positive status. In the recent era, the qualification of detected mutation is essential for the clinical purposes. Please, could the authors highlight how this data impact on the clinical results?

Response4:

We used immunohistochemical method until 2017 and with that method, we tested only 15 patients, therefore samples are not comparable.

In adults, diffuse gliomas according to new classification have been divided into three types:1) astrocytoma, IDHm; 2) oligodendroglioma, IDHm and 1p/19q-codeleted; and 3) glioblastoma, IDHwt. Until then glioblastomas were diagnosed based on the histologic findings including both IDHm and IDHwt tumors with very different biologies and prognoses. In WHO CNS5, glioblastomas will comprise only IDHwt tumors. In addition, IDHwt diffuse astrocytic tumors in adults without the histologic features of glioblastoma but having one or more of three genetic parameters (TERT promoter mutation,EGFR gene amplification, combined gain of entire chromosome 7 and loss of entire chromosome 10) will also be classified as glioblastomas. In the new classification, all IDHm diffuse astrocytic tumors are considered a single type astrocytoma, IDHm  and are graded as 2, 3, or 4. Grading of these tumors will also take into account other molecular findings such as the presence of CDKN2A/B homozygous deletion, which results in a worse prognosis and which will now have a WHO grade of 4. This separation into IDH wild type and mutant tumors is an important advancement and a key factor in the treatment, follow-up, and understanding of glial tumors.

Point5: In the results section, please, could the authors analyze clinical data at the sight of pathologic classiffication of enrolled patients( grading, histological diagnosis)?

Response 5:

We divided our patients into two groups solely according to the status of the IDH mutation because if we were to consider histology and grade, the groups would be too small for statistical processing. This would be taken into account in the continuation of the study when we have a larger number of patients.

Round 2

Reviewer 2 Report

The manuscript may be accepted in the present form

Author Response

Point1: In the introduction section, please, could the authors better explain the achronymous for TP and TRC?

Response1:

TRC encompasses two situations, and these are pseudoprogression and radiation necrosis. Pseudoprogression describes the phenomenon that, in the absence of actual tumor growth, the diameter of contrast-enhancing areas enlarges more than 25% or new lesions occur during or after therapy, mimicking tumor progression within the first three months after chemoradiation completion with subsequent improvement of MRI findings. Radiation necrosis denotes an injury of brain tissue that is related to irradiation and may occur several months or even years after radiotherapy completion. Since we investigated the late effects of radiochemotherapy, TRC was associated with radiation necrosis in our study.

Point2: In the material and methods section (lines 68 -75) please, could the authors also report the incluision criteria followed to enroll patients in this study? In addition, could the authors report if biological material was administrated according to international guideliens?

Response2:

This retrospective study included 47 patients who were treated at the Division of Radiotherapy, Institute of Oncology in Ljubljana and, on the recommendation of the multidisciplinary tumor board and in order to distinguish between TP and TRC, were referred to our Nuclear Medicine Department for 18F-FET PET imaging between April 2019 and October 2021. All patients who had undergone standard MRI were able to understand the reason for additional 18F-FET PET imaging. All patients had previously been diagnosed with adult diffuse gliomas and had a prior biopsy and radiochemotherapy according to EANO guidelines.

The study was approved by the institutional review board committee (approval number ERIDNPVO-0073/2021) and biological material was administered according to international guidelines.

Point3: In the material and methods section (lines 76-80), the authors describe the conventional technology adopted to evaluate ID1 and IDH2 mutational status. Accordingly, NGS approach is not adequately discussed. Please, could the authors report technical details (reference range, sensitivity, specificity, limit of detection) usefull to accept positive status in IDH1/2 molecular analysis?

Response3:

Next-generation sequencing (NGS) is increasingly being used in routine clinical practice, including for the diagnosis of rare entities like gliomas because it can replace multiple single-gene genomic testing technologies while requiring only one test. Gene-targeted NGS offers a cost-effective approach to simultaneously detect multiple genetic alterations with a minimal amount of DNA and with high sensitivity, which makes this method highly attractive for use in gliomas. Panels that are specifically designed for gliomas are needed for the routine diagnosis of these tumors. We use the isolation of DNA from FFPE tissue using the Maxwell RSC FFPE Plus DNA Purification kit (Promega). New generation target sequencing (NGS, Ion S5, AmpliSeq Glioma Custom Design, Thermo Fisher Scientific) at the level DNA: analysis of somatic mutations in specific target regions ("Hotspots") in 9 genes:BRAF, H3F3A, HIST1H3B, HIST1H3C, IDH1, IDH2, KRAS, NRAS, pTERT, analysis of the entire coding region of 11 genes:ACVR1, ATRX, CIC, FUBP1, EGFR, FGFR1, PIK3CA, PIK3R1, PTEN, SETD2, TP53,12 gene copy number (CNV) analysis:CDKN2A, CDKN2B, EGFR, FGFR1, MDM2, MDM4, MET, MYCN, PDGFRA, PIK3CA, PIK3R1, PTEN and chromosome level analysis: chromosome lever 1p, 10q and 19q and chromosome 7. Analysis of results using the IonReporter software package (Thermo Fisher Scientific) (reference genome hg19). Border detection (default filter) is set to 5.0% (mutation rate relative to unmutated DNA). Negative result (no mutations) does not exclude the presence of mutations, amplifications or deletions below the limit of detection.

Point 4: In the results section, the authros elucidate positive status for IDH1/2 in enrolled patients. As regards, please, could the authors evaluate if sttistically relevant differences may be observed according to the technology? In addition, it is anachronystic only consider IDH1/2 positive status. In the recent era, the qualification of detected mutation is essential for the clinical purposes. Please, could the authors highlight how this data impact on the clinical results?

Response4:

We used immunohistochemical method until 2017 and with that method, we tested only 15 patients, therefore samples are not comparable.

In adults, diffuse gliomas according to new classification have been divided into three types:1) astrocytoma, IDHm; 2) oligodendroglioma, IDHm and 1p/19q-codeleted; and 3) glioblastoma, IDHwt. Until then glioblastomas were diagnosed based on the histologic findings including both IDHm and IDHwt tumors with very different biologies and prognoses. In WHO CNS5, glioblastomas will comprise only IDHwt tumors. In addition, IDHwt diffuse astrocytic tumors in adults without the histologic features of glioblastoma but having one or more of three genetic parameters (TERT promoter mutation,EGFR gene amplification, combined gain of entire chromosome 7 and loss of entire chromosome 10) will also be classified as glioblastomas. In the new classification, all IDHm diffuse astrocytic tumors are considered a single type astrocytoma, IDHm and are graded as 2, 3, or 4. Grading of these tumors will also take into account other molecular findings such as the presence of CDKN2A/B homozygous deletion, which results in a worse prognosis and which will now have a WHO grade of 4. This separation into IDH wild type and mutant tumors is an important advancement and a key factor in the treatment, follow-up, and understanding of glial tumors.

Point5: In the results section, please, could the authors analyze clinical data at the sight of pathologic classiffication of enrolled patients( grading, histological diagnosis)?

Response 5:

We divided our patients into two groups solely according to the status of the IDH mutation because if we were to consider histology and grade, the groups would be too small for statistical processing. This would be taken into account in the continuation of the study when we have a larger number of patients.

  ( ) (x) ( ) ( )

  (x) ( ) ( ) ( )
    ( ) (x) ( )